Potential Arabidopsis thaliana glucosinolate genes identified from the co-expression modules using graph clustering approach

Harun Sarahani 1
Afiqah-Aleng Nor 2
Karim Mohammad Bozlul 3
Altaf Ul Amin Md 3
Kanaya Shigehiko 3
http://orcid.org/0000-0002-5386-7260 Mohamed-Hussein Zeti-Azura 1 4 zeti.hussein@ukm.edu.my
1 Centre for Bioinformatics Research, Institute of Systems Biology (INBIOSIS), Universiti Kebangsaan Malaysia , UKM Bangi, Selangor , Malaysia
2 Institute of Marine Biotechnology, Universiti Malaysia Terengganu , Kuala Nerus, Terengganu , Malaysia
3 Graduate School of Science and Technology & NAIST Data Science Center, Nara Institute of Science and Technology , Nara , Japan
4 Department of Applied Physics, Faculty of Science and Technology, Universiti Kebangsaan Malaysia , UKM Bangi, Selangor , Malaysia
Beemster Gerrit
Electronic publication date: 2021 Aug 4
Publication date: 2021
Volume: 9
Electronic Location ID: e11876
Received 2021 May 6; Accepted 2021 Jul 6
Copyright: © 2021 Harun et al.
Copyright year: 2021
Copyright holder: Harun et al.
License: This is an open access article distributed under the terms of the Creative Commons Attribution License, which permits unrestricted use, distribution, reproduction and adaptation in any medium and for any purpose provided that it is properly attributed. For attribution, the original author(s), title, publication source (PeerJ) and either DOI or URL of the article must be cited.
License URL: https://creativecommons.org/licenses/by/4.0/

Keywords: Secondary metabolites, Nitrogen-containing compounds, Aliphatic glucosinolates, Indolic glucosinolates, Graph clustering, Gene network analysis

Funding: Malaysian Ministry of Higher Education ERGS/1/2013/STG07/UKM/02/3 JASSO This work was supported by the Malaysian Ministry of Higher Education (ERGS/1/2013/STG07/UKM/02/3) awarded to Zeti-Azura Mohamed-Hussein. Sarahani Harun is funded by JASSO for her attachment at NAIST to perform this experiment. The funders had no role in study design, data collection and analysis, decision to publish, or preparation of the manuscript.

==============================
Background

Glucosinolates (GSLs) are plant secondary metabolites that contain nitrogen-containing compounds. They are important in the plant defense system and known to provide protection against cancer in humans. Currently, increasing the amount of data generated from various omics technologies serves as a hotspot for new gene discovery. However, sometimes sequence similarity searching approach is not sufficiently effective to find these genes; hence, we adapted a network clustering approach to search for potential GSLs genes from the Arabidopsis thaliana co-expression dataset.

Methods

We used known GSL genes to construct a comprehensive GSL co-expression network. This network was analyzed with the DPClusOST algorithm using a density of 0.5. 0.6. 0.7, 0.8, and 0.9. Generating clusters were evaluated using Fisher’s exact test to identify GSL gene co-expression clusters. A significance score (SScore) was calculated for each gene based on the generated p-value of Fisher’s exact test. SScore was used to perform a receiver operating characteristic (ROC) study to classify possible GSL genes using the ROCR package. ROCR was used in determining the AUC that measured the suitable density value of the cluster for further analysis. Finally, pathway enrichment analysis was conducted using ClueGO to identify significant pathways associated with the GSL clusters.

Results

The density value of 0.8 showed the highest area under the curve (AUC) leading to the selection of thirteen potential GSL genes from the top six significant clusters that include IMDH3, MVP1, T19K24.17, MRSA2, SIR, ASP4, MTO1, At1g21440, HMT3, At3g47420, PS1, SAL1, and At3g14220. A total of Four potential genes (MTO1, SIR, SAL1, and IMDH3) were identified from the pathway enrichment analysis on the significant clusters. These genes are directly related to GSL-associated pathways such as sulfur metabolism and valine, leucine, and isoleucine biosynthesis. This approach demonstrates the ability of the network clustering approach in identifying potential GSL genes which cannot be found from the standard similarity search.

Introduction

Plant secondary metabolites are divided into three chemically distinct classes: terpenes, phenolics, and nitrogen-containing compounds (Taiz & Zeiger, 2010). Alkaloids, cyanogenic glycosides, non-protein amino acids (NPAAs), and glucosinolates are the examples of nitrogen-containing compounds (Wink, 2015). Glucosinolates (GSLs) are known for protecting plants against invading pests and pathogens as well as preventing cancer in humans (Tang et al., 2010; Lai et al., 2010; Frerigmann et al., 2016; Megna et al., 2016; Piślewska-Bednarek et al., 2017). GSLs are found in the Brassicaceae family known as cruciferous vegetables, consisting of broccoli, cabbage, cauliflower, kale, mustard and cress (Herr & Büchler, 2010), and in the model plant, Arabidopsis thaliana (Redovniković et al., 2008). In 2001, 34 GSLs were identified from the leaves and seeds of 39 different Arabidopsis ecotypes, most of which were chain-elongated products originated from methionine (Met) (Kliebenstein et al., 2001).

GSLs are one of the most studied secondary metabolites since the beginning of 2000 until now (Sønderby, Geu-flores & Halkier, 2010; Burow et al., 2015). Interests in plant GSLs rise in recent years because of their importance in plant defense and cancer preventive agent (Tang et al., 2011; Megna et al., 2016), and other beneficial effects such as providing regulatory function inflammation, stress responses, antioxidant and antimicrobial properties (Bischoff, 2016). GSLs are amino acid-derived compounds divided into three main categories: aliphatic GSLs, derived from Ala, Leu, Ile, Val, and Met; benzyl GSLs, derived from Phe or Tyr; and indolic GSLs, derived from Trp. The GSL pathway consists of several genes that encode for transcription factors, transporters and enzymes involved in the biosynthesis of GSLs. The genes are essential in the side-chain elongation, core structure synthesis, side-chain modification, as well as GSL degradation (Agerbirk & Olsen, 2012; Blažević et al., 2019).

In recent years, the identification of genes involved in the GSL biosynthesis has been extensively investigated using gene co-expression data (Gachon et al., 2005; Hirai et al., 2005, 2007; Knill et al., 2008; Sawada et al., 2009a). This approach is used to identify novel genes that encode for transcription factors (TFs) and enzymes involved in the GSL biosynthesis (Hirai et al., 2005, 2007; Knill et al., 2008; Sawada et al., 2009a). Hirai et al. (2007) have proven the association between MYB28 and MYB29 with aliphatic GSL biosynthesis; both genes were previously unknown to be responsible in encoding R2R3-Myb TFs. Their analysis has also shown the co-expression of TFs with several known aliphatic GSLs: cytochrome P450 79F1 (CYP79F1), cytochrome P450 79F2 (CYP79F2), methylthioalkylmalate synthase 1 (MAM1) and methylthioalkylmalate synthase 3 (MAM3) (Hirai et al., 2007). Meanwhile, Sawada et al. (2009b) have identified bile acid transporter 5 (BAT5) to co-express with the aliphatic GSL genes involved in chain elongation.

Recently (Harun et al., 2020) have compiled a total of 113 known GSL genes with experimental evidences from the published research conducted for the last 20 years. They classified the genes according to their annotation and grouped them into TFs, biosynthetic genes, and protein transporters. These genes are used as bait genes to find “additional/missing genes” from the co-expression modules in order to identify more novel GSL genes. Thus, a computational pipeline of biological network approach can accelerate the finding of these genes. Previous studies demonstrated the application of the graph clustering approach in the protein-protein interaction (PPI) network followed by the Fisher’s exact test to identify disease clusters in the inflammatory bowel disease (IBD) (Eguchi et al., 2018) and polycystic ovarian syndrome (PCOS) (Afiqah-Aleng et al., 2020). The identification of the novel disease genes and related pathways is able to discover disease-associated genes that are linked with other diseases as well. Hence, understanding the biological components involved would lead to additional insight into the mechanism of both diseases that lead to effective treatment in the future (Eguchi et al., 2018; Afiqah-Aleng et al., 2020). In Oryza sativa, a clustering approach of a PPI network was able to elucidate a molecular mechanism of nitrate that regulates nitrite reductase, ferredoxin-NADP reductase, and ferredoxin. These three components are associated with flowering time that led to a novel contribution of nitrate signaling in light and dark environment (Pathak et al., 2020). In this article, we describe the computational pipeline that involves the use of graph clustering for novel genes annotation in plants. In this study, we used the abovementioned approach to search for potential GSLs biosynthetic genes which were previously unknown genes.

Materials & Methods

Data collection and co-expression network construction

Before this study, only 46 known GSL genes were identified by Sønderby, Geu-Flores & Halkier (2010). Following that, we performed a comprehensive literature and pathway databases search in KEGG (http://www.genome.jp/kegg/) (Kanehisa et al., 2016) and AraCyc (https://www.arabidopsis.org/biocyc/) (Mueller, Zhang & Rhee, 2003) to identify more known GSL genes using keywords, such as glucosinolate AND Arabidopsis, in journals published in 2020 (Harun et al., 2020). The list of updated GSL genes were also added in our manually curated sulfur compound database, SuCComBase (http://plant-scc.org/) (Harun et al., 2019).

We utilized ATTED-II database (http://atted.jp/) (Aoki et al., 2016), a database of co-expressed genes, initially involving Arabidopsis and rice, to identify candidate genes co-expressed with known GSLs. In ATTED, the Arabidopsis RNAseq and microarray data from ArrayExpress (Rustici et al., 2013) covered 94% and 76% of the Arabidopsis protein-encoding genes, respectively. In this study, we used three additional co-expression tools: AraNet v2 (Lee et al., 2014), GeneMANIA (Lee et al., 2014), and STRING (Szklarczyk et al., 2015, 2017, 2019). The co-expression data in AraNet covers 83.5% of the Arabidopsis coding genome from the Gene Expression Omnibus (GEO) database involving 1,261 microarray experiments (Barrett et al., 2013).

We used known GSL genes as a query against three co-expression tools in an effort to search for “additional” genes that were co-expressed with them. We defined “additional” genes as potential GSL genes that will be critically and systematically assessed using cluster and pathway enrichment analysis before mapping them on the GSL biosynthesis pathway. Known GSL genes were used as a query against the co-expression tools, including ATTED-II version 10.1 (http://atted.jp/) (Aoki et al., 2016), AraNet v2 (Lee et al., 2014), GeneMANIA (Warde-Farley et al., 2010; Montojo et al., 2014) and STRING (Szklarczyk et al., 2015, 2017, 2019). A total of Four gene networks were combined into a single gene co-expression network using Cytoscape 3.7.1 (Shannon et al., 2003). The steps involved in data establishment and gene co-expression network construction are shown in Fig. 1.

Figure 1 Step-by-step procedure to identify potential GSL genes involved in the GSL biosynthetic pathway.

The list of plant transcription factors (TFs) in A. thaliana was downloaded from the Plant Transcription Factor Database v5.0 (PlantTFDB 5.0; http://planttfdb.cbi.pku.edu.cn) (Jin et al., 2016). The information generated from PlantTFDB will add Biological information of each potential GSL gene was added in this study using various databases, such as UniProt (https://www.uniprot.org/) (Bateman et al., 2017) and TAIR (https://www.arabidopsis.org/index.jsp) (Lamesch et al., 2012).

Calculating clusters

DPClusOST (Bozlul Karim, Wakamatsu & Altaf-Ul-Amin, 2017), an option in DPClusO algorithm (Altaf-Ul-Amin et al., 2006; Altaf-Ul-Amin, Wada & Kanaya, 2012) was used to generate clusters in order to identify densely connected regions from a gene network using a graphical interface. The clustering algorithm generates overlapping clusters that influenced several biological processes related to GSL metabolism. DPClusO is used for an undirected graph consisting of a finite set of nodes N and a finite set of edges E. In this algorithm, two critical parameters are introduced: density dk and cluster property cpnk. Density dk of cluster k refers to the ratio of the number of actual cluster edges (|Ek|) and the maximum possible number of cluster edges (|Ek|max). Detailed information on this algorithm was described in previous studies by Eguchi et al. (2018) and Afiqah-Aleng et al. (2020). The cluster property of node n with respect to cluster k is represented by the following equation:

(1) cpnk=|Enk|dk×|Nk|

Nk refers to the number of nodes in cluster k. Enk is the total number of edges connecting the node n with nodes of cluster k.

Fisher’s exact test

Fisher’s exact test (Fisher, 1992) was used to evaluate known GSL gene enrichment clusters. It is a statistical test used in the analysis of 2 × 2 contingency tables (Fisher, 1922, 1992). The introduced values of a, b, c and d are shown in Table 1. In order to identify the best set of clusters, a calculation to obtain the average significance of a cluster set was introduced. Fisher’s exact test p-values were calculated to assess GSL genes enrichment in each of the identified clusters.

Table 1 The contingency table prepared in this study to calculate known GSL gene enrichment clusters.

	GSL genes	Non-GSL genes	
In cluster	a	b	
Not in cluster	c	d	
	a+c	b+d	
Note:

aHere n refer to the total number of genes in the gene network.

SScore and ROC analysis

The prediction confidence of potential GSL genes was calculated for each gene depending on the p-value of the generated clusters using a significance score (SScore). A similar approach was used in the protein-protein interaction network on human diseases as described by Eguchi et al. (2018) and Afiqah-Aleng et al. (2020). The formula for SScore was SScore = −log (p-value). Since DPClusO produced overlapping clusters, the lowest p-value of a gene was used to measure SScore. A gene can belong to more than one cluster and equate to more than one p-value. Next is the receiver operating characteristic (ROC) analysis that was conducted to identify the potential GSL genes by calculating the power of SScore (Metz, 1978; Davis & Goadrich, 2006). True Positive Rate (TPR) and False Positive Rate (FPR) were calculated using a series of threshold (th) SScore in this study. The fraction of true positive predictions in all positive data is TPR, and the fraction of false positive predictions in all negative data is FPR. The following equations were used to calculate TPR and FPR:

(2) TPR=TPTP+FN

(3) FPR=FPFP+TN

Based on the listed equations above, true positive (TP), false positive (FP), true negative TN, and false negative (FN) were known as the number of GSL genes with SScore≥th, number of non-GSL genes with SScore≥th, number of non-GSL genes having SScore<th and number of GSL genes having SScore<th, respectively. The Area Under the ROC Curve (AUC) study was used to evaluate the efficiency of SScore in identifying potential GSL genes. ROCR (Sing et al., 2005), a R package, was used to calculate the AUC in this study.

Pathway enrichment analysis

To evaluate the biological role of the clusters, pathway enrichment analysis was performed on the potential GSL genes and GSL clusters against pathway databases, including Kyoto Encylopedia of Genes and Genomes (Kanehisa et al., 2017) using ClueGO/CluePedia (Bindea et al., 2009) apps in Cytoscape. The false discovery rate of each pathway was calculated using a hypergeometric test with Bonferroni correction to determine its importance. To define the relation between pathways, a Kappa score of 0.5 was chosen. The overview of each step taken in the gene network clustering approach, statistical analysis on the significant clusters, and pathway enrichment analysis are shown in Fig. 1.

Results

Identification of GSL genes and co-expression network construction

All information on the genes encoding proteins involved in GSL was extracted using the sources shown in Fig. 1. Unlike Kyoto Encyclopedia of Genes and Genomes (KEGG), there are 12 GSL pathway derivatives in AraCyc: aliphatic GSL biosynthesis (side-chain elongation cycle), GSL biosynthesis from homomethionine, GSL biosynthesis from dihomomethionine, GSL biosynthesis from trihomomethionine, GSL biosynthesis from tetrahomomethionine, GSL biosynthesis from pentahomomethionine, GSL biosynthesis from hexahomomethionine, GSL biosynthesis from phenylalanine, GSL biosynthesis from tryptophan, GSL breakdown, indole GSL breakdown (active in intact plant cell) and indole GSL breakdown (insect chewing induced). Finally, a total of 113 known GSL genes (experimentally verified GSL genes) were used throughout this study.

The 113 known GSL genes were used as bait genes to query the whole transcriptomics data from four co-expression network tools (ATTED, GeneMANIA, STRING, and AraNet v2). Figure 2 shows the interaction between 112 GSL genes with 158 interacting partners, generating 5,554 edges. This network was constructed from four gene co-expression networks that produced 161 nodes and 4,108 edges from GeneMANIA; 88 nodes and 355 edges from AraNet; 161 nodes and 2325 edges from STRING, and 197 nodes and 370 edges from ATTED. These individual gene co-expression networks were merged using Cytoscape to produce an integrated gene co-expression network consisting of 270 nodes and 5,554 edges (Fig. 2).

Figure 2 A gene co-expression network that consists of 250 nodes and 5,554 edges.

The gene co-expression network (Fig. 2) consists of various functional groups based on the mechanism in GSL biosynthesis. AOP2 (alkenyl hydroxyalkyl producing 2) was the only known GSL gene that was not found in this gene network. The absence of AOP2 in the gene network might due to lack of co-expression data that link the bait gene with other genes in the co-expression databases. The transcriptional components in GSL biosynthesis were divided into their respective GSL regulatory mechanism, such as activator, repressor, and mediator. The GSL regulatory network would affect several GSL biosynthetic pathways that can be grouped into side-chain elongation, core structure synthesis, co-substrate pathways, and side chain modification (Harun et al., 2020). GSL degradation refers to the formation of activated GSL products that are known to confer protection in plants against the biotic, and abiotic stresses (Halkier & Gershenzon, 2006; Liu et al., 2020). There are also five known GSL transporters in the gene network: BAT5, SULTR1;1, SULTR1;2, GTR1, and GTR2. In Fig. 2, the 158 additional genes that are defined as interacting partners with the known GSL are characterized into seven potential GSL transcription factors (TFs), and 151 potential biosynthetic GSL genes that will be further analyzed in this study.

Gene co-expression clusters analysis

Once the gene co-expression network was constructed, the present clusters in the network were determined using the DPClusOST algorithm. DPClusOST extracts highly interconnected region that perform a similar biological process. We hypothesize that co-exist genes with known GSL genes in the same statistically significant clusters can be used to predict potential GSL genes. These co-exist genes are the additional genes in the gene co-expression network clustered with known GSL genes. A total of five sets of clusters were generated using density values of 0.5, 0.6, 0.7, 0.8, and 0.9 with cp value of 0.5 (Table 2). The density dk of any cluster k refers to the ratio of the number of edges in the cluster (|Ek|) and the maximum possible number of cluster edges (|Ek|max). Clusters generated from different density values produced distinctive cluster characteristics, namely the number of clusters, the maximum size of the cluster, and the average cluster size. Smaller density values resulted in greater cluster sizes and fewer clusters, as expected. As for cp value, 0.5 is the default and recommended value and has been used in previous studies (Eguchi et al., 2018; Karim et al., 2020).

Table 2 Cluster properties of different input densities using DPClusO algorithm.

Density	Number of cluster	maxsize	avgsize	
0.5	95	125	31.41	
0.6	152	102	17.13	
0.7	163	82	62.96	
0.8	186	64	42.44	
0.9	213	47	15.08	

From the five different input densities, DPClusOST generated five sets of clusters. To determine which set of clusters for further analysis, we performed a receiver operating characteristic (ROC) analysis. First, Fisher’s exact test p-values were calculated to assess GSL genes enrichment in each of the identified clusters. Then we assigned the significance score (SScore), to each gene, based on the p-values of the clusters to which they belong. Next, we created five ROC curves by utilizing the SScore corresponding to the five sets of clusters. The AUC of five ROC curves is shown in Fig. 3. The maximum AUC was 0.87, generated from the density value of 0.8. The potential GSL genes found within the statistically significant clusters of the set corresponding to density 0.8 were selected as potential GSL genes (Table S1).

Figure 3 The calculated area under curve (AUC) of five clusters generated using DPClusO.

The density value of 0.8 has the highest AUC, followed by clusters generated from density value of 0.9, 0.7, 0.6, and 0.5.

A total of 148 significant clusters with a density value of 0.8 (p-value < 0.05) was identified, with the potential GSL genes found within the statistically significant cluster being considered significant and analyzed further in this study. The overall result of the 148 significant clusters are shown in Table S2. Table 3 shows a list of potential GSL genes identified from the selected highly significant clusters. Based on Table S2, the top six highly significant clusters (Cluster 121, Cluster 131, Cluster 127, Cluster 125, Cluster 129, and Cluster 128) were chosen for further analysis. The genes in the light blue nodes referred to the potential GSL genes in our study.

Table 3 List of potential GSL genes from selected highly significant clusters.

Cluster number	Cluster size	Potential GSL genes (number)	Cluster	p-value	
Cluster 121	51	IMDH3, MVP1, T19K24.17, MRSA2 (4)		5.79E−17	
Cluster 131	34	MRSA2 (1)		2.27E−13	
Cluster 127	44	T19K24.17, SIR, MRSA2, ASP4, CGS1, At1g21440 (6)		2.75E−11	
Cluster 125	46	T19K24.17, SIR, MRSA2, ASP4, CGS1, At1g21440, HMT3 (7)		4.65E−11	
Cluster 129	41	At3g47420, PS1, SIR, T19K24.17, SAL1, CGS1 (6)		4.77E−10	
Cluster 128	42	At3g14220, T19K24.17, SIR, MRSA2, ASP4, CGS1, At1g21440 (7)		1.93E−09	

Based on Table 3, a total of thirteen potential GSL genes were identified from the top six significant clusters: IMDH3, MVP1, T19K24.17, MRSA2, SIR, ASP4, MTO1, At1g21440, HMT3, At3g47420, PS1, SAL1, and At3g14220. Each cluster contained known GSL genes with functions that included transcription factors and other related GSL biological processes, such as side-chain elongation, core structure synthesis, side-chain modification, and GSL degradation. These genes were also grouped into aliphatic and indolic GSL genes depending on their involvement in the type of GSL being produced. There were also a group of gene-encoding enzymes involved in GSL degradation and a GSL transporter (BAT5).

Pathway enrichment analysis

KEGG pathway enrichment was identified from the selected top six significant clusters and five significant pathways, i.e., selenocompound metabolism, sulfur metabolism, tryptophan metabolism, valine, leucine, and isoleucine biosynthesis, and GSL biosynthesis (Fig. 4) were selected. Pink nodes denote the known GSL genes that encode for enzymes, TFs, and protein transporters whilst blue nodes denote the potential GSL genes might involve in GSL biosynthesis.

Figure 4 Visualization of KEGG pathway enrichment using ClueGO/CluePedia apps from Cytoscape.

The enrichment shows only significant pathways (p-value ≤ 0.05).

Discussion

In this study, we propose a combinatorial approach as an alternative way to identify possible GSL genes as compared to using the traditional sequence-based searching. Our method demonstrates that graph clustering analysis combined with Fisher’s exact test and ROC analysis is able to classify possible genes that aren’t identified or were missed using the standard sequence-based searching. We believe it is critical to provide an alternative technique that is able to search and classify the sequences without relying solely on the sequence searching technique. DPClusOST is a clustering algorithm that discovers and classifies strongly interconnected regions in a large network between core nodes, or high connectivity nodes and peripheral nodes, or low connectivity nodes to indicate biological significance in a cell. A similar approach was used in analysing protein-protein interaction network in inflammatory bowel disease (IBD) (Eguchi et al., 2018) and polycystic ovarian syndrome (PCOS) (Afiqah-Aleng et al., 2020). DPClusOST creates overlapping clusters depending on a gene’s multifunctionality, resulting in a high probability of a gene being present in multiple clusters (Altaf-Ul-Amin, Wada & Kanaya, 2012; Bozlul Karim, Wakamatsu & Altaf-Ul-Amin, 2017). The DPClusOST algorithm extracts highly interconnected region that perform similar biological process. Thus, integrating this algorithm in our pipeline demonstrate its ability to suggest that the existence of genes in the same statistically significant clusters with the known GSL genes can be used to predict potential GSL genes.

DPClusOST algorithm produced five sets of clusters generated from five density values. Fisher’s exact test p-value and SScore with the AUC value were used to assess the cluster. The maximum AUC was identified from the clusters generated from density value of 0.8 producing 127 potential GSL genes (Table S1). One of the potential GSL gene, ADAP (ARIA-interacting double AP2-domain protein) was identified from the clustering approach and Fisher’s exact test. Further functional analysis was performed using chimeric repressor gene silencing technology (CRES-T) and gene expression analysis using qPCR. The over-expression of downstream aliphatic GSL genes (UGT74C1 and IPMI1) in the ADAP-SRDX line indicated the possibility of ADAP as a negative regulator in aliphatic GSL biosynthesis via a feedback mechanism (Harun et al., 2021). A total of thirteen potential genes (Table 3) were identified from the top six significant analysis that is known as IMDH3, MVP1, T19K24.17, MRSA2, SIR, ASP4, MTO1, At1g21440, HMT3, At3g47420, PS1, SAL1, and At3g14220. Interestingly, MRSA2 and At1g21440 were found to be co-expressed with known aliphatic GSL biosynthesis in Arabidopsis in a large-scale analysis of plant gene co-expression network of specialized metabolic pathways. MRSA2 is peptide methionine sulfoxide reductase protein whereas At1g21440 is a phosphoenolpyruvate carboxylase family protein where they were both not known to be involved in the aliphatic GSL biosynthesis (Wisecaver et al., 2017).

Pathway enrichment analysis using ClueGO/CluePedia (Bindea et al., 2009) apps from Cytoscape (Shannon et al., 2003) was used to interpret the association of the potential GSL genes and the selected significant clusters in Table 3. Enrichment analysis is used to map known biological functions of the generated clusters that were extracted from pathway databases such as KEGG (de Anda-Jáuregui, 2019). Figure 4 shows map of pathway enrichment-merged pathway that contains five enriched clusters that are related with GSL biosynthesis, i.e., sulfur metabolism, tryptophan metabolism, and valine, leucine and isoleucine biosynthesis. Since GSLs have at least two sulfur atoms in their main structure, and aliphatic GSLs may have additional sulfur in their side chains, sulfur metabolism might have an importance to GSL biosynthesis (Falk, Tokuhisa & Gershenzon, 2007). Sulfur content in GSLs indicates that this compound is critical in GSL biosynthesis. Several metabolomic and transcriptomic studies reported a significant reduction in GSL accumulation under sulfur deficiency environment, suggesting the role of sulfur in GSL biosynthesis (Nikiforova et al., 2003; Hirai et al., 2003; Aarabi et al., 2016). SULTR1; 1 and SULTR1; 2 in Arabidopsis roots play a role as sulfate transporters and their expression was increased in sulfur-limitation Arabidopsis (Koprivova & Kopriva, 2014; Morikawa-Ichinose et al., 2019). The molecular components involving sulfate and GSL transport machinery is more complex in Brassica crops, and requires an in-depth understanding on the GSL mechanism. From Fig. 4, several GSL genes are grouped in GSL core-substrate pathways, such as 5′-adenylylsulfate reductases (APR1 and APR2), adenylyl-sulfate kinases (APK1 and APK2), and ATP sulfurylases (APS1 and APS3) and they were identified to be involved in both aliphatic and indolic GSL biosynthesis and linked to sulfate assimilation (Yatusevich et al., 2010a; Harun et al., 2020).

There are three potential GSL genes directly linked to sulfur metabolism (Fig. 4): cystathionine gamma-synthetase 1 (MTO1), sulfite reductase (SIR), and SAL1 phosphatase (SAL1). CGS is the key enzyme in methionine biosynthesis located in the chloroplast (Takahashi et al., 2011). SIR has been previously identified in sulfate assimilation that catalyzes the production of sulfide. Sulfide undergoes a cysteine biosynthesis as well as other sulfur-containing compounds, such as GSLs (Miao et al., 2016). Previous study showed significant increase in the SIR expression in Arabidopsis plants with overexpressed indolic GSL TFs (MYB51, MYB122, and MYB34) and aliphatic GSL TF (MYB28) (Yatusevich et al., 2010b). SAL1 is a bifunctional enzyme that regulates the activities of 3′(2′),5′-bisphosphate nucleotidase and inositol polyphosphate 1-phosphatase (Quintero, Garciadeblás & Rodríguez-Navarro, 1996). Ishiga et al. (2017) reported a reduced level of aliphatic GSLs production in the sal1 mutants compared to the wild-type Col-0 in response to pathogen. They also showed the genes in both salicylic acid (SA) and jasmonic acid (JA) pathways were downregulated in sal1, suggesting the involvement of SAL1 in plant immunity (Ishiga et al., 2017).

In GSL biosynthesis, several groups of GSLs differ from their corresponding precursors. Collectively, there are three GSL groups: aliphatic GSLs produced from methionine, alanine, leucine, isoleucine or valine; indolic GSLs produced from tryptophan; and benzyl GSLs produced from phenylalanine or tyrosine (Barba et al., 2016; Seo & Kim, 2017; Harun et al., 2020). However, the aliphatic GSL biosynthesis also needs another crucial step, which is the side-chain elongation in the chloroplast, followed by core structure synthesis in the cytoplasm (Sønderby, Geu-flores & Halkier, 2010; Borpatragohain et al., 2016). Based on Fig. 4, valine, leucine and isoleucine biosynthesis are the most enriched among the top six clusters generated in this study. Several GSL biosynthetic genes (BCAT3, IPMI1, IPMI2, and IIL1) are involved in the side-chain elongation process in the biosynthesis of valine, leucine and isoleucine. These genes are known as aliphatic GSL biosynthetic genes. The isopropylmalate dehydrogenases have been reported in side-chain GSL biosynthesis involving oxidative decarboxylation that produces a chain-elongated 2-oxo acid GSL. In this study, isopropylmalate dehydrogenase 3 (IMDH3) was also identified as a potential GSL gene which directly involved in the biosynthesis of valine, leucine and isoleucine. Previous T-DNA mutant studies on Arabidopsis IMDH1, IMDH2, and IMDH3 showed a significantly decreased level of aliphatic GSLs and leucine in IMDH1, suggesting a clear role of IMDH1 in catalyzing the oxidative decarboxylation step of aliphatic GSL biosynthesis (He et al., 2011b; Lee, Nwumeh & Jez, 2016) and double mutant of both IMDH2 and IMDH3 showed alteration in pollen and embryo sac growth, suggesting their correlation between leucine biosynthesis and the gametophyte formation in Arabidopsis (He et al., 2011a).

Tryptophan metabolism pathway was found in the gene network as shown in Fig. 4 where all genes directly linked to tryptophan metabolism are involved in indolic GSLs (Seo & Kim, 2017; Harun et al., 2020). The genes can be grouped based on their function in the indolic GSL biosynthesis: GSL core structure synthesis (CYP79B2, CYP79B3, CYP83B1, SUR1, and UGT74B1), as well as GSL degradation (TGG1, TGG2, and TGG4). CYP79B2 and CYP79B3 are P450 enzymes that catalyze the production of aldoximes from the tryptophan derivatives (Mikkelsen & Halkier, 2003). Next, another member of the P450 family, CYP83B1 catalyzes the oxidation of aldoximes into nitrile oxides, which is another crucial step in the GSL core structure synthesis (Naur et al., 2003). The myrosinase enzymes (TGG, EC 3.2.1.147) facilitates the production of active GSLs (bioactive isothiocyanates, nitriles, thiocyanates, and epithionitriles) in damaged plant cells during pest attacks or food preparation. Such activated GSL products have protective roles in plants against the biotic and abiotic stresses (Halkier & Gershenzon, 2006; Liu et al., 2020). The variations of the products are based on the GSLs side chain composition and the involvement of the myrosinase interacting proteins in the GSL-myrosinase system (Wittstock et al., 2016; Chhajed et al., 2019).

Lastly, the GSL biosynthesis linked with both aliphatic and indolic GSLs was also enriched in the gene network (Fig. 4). The aliphatic side-chain elongation genes were MAM1, MAM3, BCAT3, BCAT4, IPMI1, IPMI2, and IIL1. Methylthioalkylmalate synthase 1 (MAM1) is among the earliest aliphatic GSL genes identified in 2001. It is located in the GSL-ELONG locus, known to control the biosynthesis of GSL (Kroymann et al., 2001). Another gene that controls aliphatic side-chain elongation GSL is branched-chain aminotransferase 4 (BCAT4), and in vivo analysis of BCAT4 knockout plants showed significantly reduced Met-derived aliphatic GSL production (Schuster et al., 2006). Another group of enzyme is involved in the core structure synthesis (CYP79F1, CYP83A1, CYP79A2, SUR1, UGT74B1, SOT16, SOT17, and SOT18). In the aliphatic GSL core structure synthesis, CYP79F1 converts all chain-elongated methionine derivatives into aldoximes (Hansen et al., 2001). The aldoximes are then oxidized by CYP83A1 into activated aci-nitro compounds (Naur et al., 2003) CYP79A2 is specifically involved in benzyl GSL production as identified from the engineering of benzyl GSL pathway in Nicotiana benthamiana (Wittstock & Halkier, 2000; Geu-Flores et al., 2009). SUR1 and UGT74B1 are involved in aliphatic and indolic GSL biosynthesis via C-S lyase reaction and glycosylation, respectively. The end-product, desulfoGSLs would undergo sulfonation via the sulfotransferases (SOT16, SOT17, and SOT18), producing the GSL core structure in the cytosol (Piotrowski et al., 2004; Harun et al., 2020).

All enriched pathways identified in the gene network are known to be involved in GSL biosynthesis and metabolism. However, there are several known TFs and biosynthetic genes that are previously not linked with the terms in the pathway enrichment due to the usage of KEGG pathway in the ClueGO apps. We realized that side-chain modification was unavailable in the KEGG database even though side-chain modification pathway is one of the crucial step in GSL bioynthesis where the production of side chains would determine the biological functions of the activated GSL end products (Harun et al., 2020). The flavin monoxygenases (FMOGS-OX1, FMOGS-OX2, FMOGS-OX3, and FMOGS-OX5) are the side-chain modification enzymes that catalyze S-oxygenation process of methylthioalkyl GSL to methylsulfinylalkyl GSL. This process influences further modifications of GSL core structure that later produces the final GSL hydrolysis products in GSL biosynthesis (Hansen et al., 2007; Kong et al., 2016). Other missing components in the metabolic pathway database are the regulatory genes that encode for TFs and transporter proteins, both are crucial in the multi-component pathways like GSL biosynthesis. Thus, by referring to the latest articles and databases such as our in-house database, SuCCombase, the latest information of GSL components could improve the constructed GSL gene network.

The discovery of potential GSL genes (MTO1, SIR, SAL1, and IMDH3) from the pathway enrichment suggest their contribution in the aliphatic GSL biosynthesis based on their co-expression with known aliphatic GSL biosynthetic genes obtained from our proposed approach. Several potential genes unlinked to any enriched GSL pathway but might be involved in GSL biosynthesis: MVP1, T19K24.17, MRSA2, ASP4, At1g21440, HMT3, At3g47420, PS1, and At3g14220 were identified. These genes were not mentioned in any known GSL pathway in the KEGG PATHWAY database; however, they were observed in the significantly enriched clusters obtained from the calculated AUC value. This finding is worth for a molecular validation e.g., mutant studies, functional studies of downstream genes, and targeted metabolomics approach in order to prove their involvement in the GSL biosynthesis.

Conclusions

Previously, we have successfully identified a novel GSL gene ADAP (Table S1) in the GSL biosynthesis via the similar approach and carried out an experimental validation of its involvement in the biosynthesis (Harun et al., 2021). Thirteen potential GSL genes from the top six significant clusters: IMDH3, MVP1, T19K24.17, MRSA2, SIR, ASP4, MTO1, At1g21440, HMT3, At3g47420, PS1, SAL1, and At3g14220 were identified from the GSL enriched clusters. Both MRSA2 and At1g21440 were the identified co-expressed genes in an aliphatic GSL co-expression network conducted in a previous study giving a high possibility of these genes as the potential GSL genes. Pathway enrichment analysis show direct involvement of four potential genes (MTO1, SIR, SAL1, and IMDH3) in the GSL biosynthesis-related pathways; sulfur metabolism and valine, leucine and isoleucine biosynthesis. This work demonstrated the application of network biology approach in the identification of missing genes and their related pathways. The combinatorial approach using graph clustering, Fisher’s exact test, and ROC analysis on the constructed network biology can be used as an alternative technique to search for missing genes that cannot be found using the traditional sequence-based searching approach. This computational pipeline will benefit the scientific community in search for valuable information in the new gene discovery efforts. Furthermore, accurate knowledge on these genes is beneficial to plant scientists in the creation of genetic resources for crop improvement.

Supplemental Information

Supplemental Information 1 127 potential GSL genes with a density value of 0.8 (p-value < 0.05).

Click here for additional data file.

Supplemental Information 2 148 significant clusters with a density value of 0.8 (p-value < 0.05).

Click here for additional data file.

We thank the Centre for Bioinformatics Research (CBR), Institute of Systems Biology (INBIOSIS), Universiti Kebangsaan Malaysia and Computational Systems Biology Lab, Nara Institute of Science and Technology (NAIST) for the computational facilities.

Additional Information and Declarations

Competing Interests

Author Contributions

Data Availability

The authors declare that they have no competing interests.

Sarahani Harun conceived and designed the experiments, performed the experiments, analyzed the data, prepared figures and/or tables, authored or reviewed drafts of the paper, and approved the final draft.

Nor Afiqah–Aleng performed the experiments, analyzed the data, prepared figures and/or tables, and approved the final draft.

Mohammad Bozlul Karim performed the experiments, analyzed the data, authored or reviewed drafts of the paper, and approved the final draft.

Md Altaf Ul Amin conceived and designed the experiments, authored or reviewed drafts of the paper, and approved the final draft.

Shigehiko Kanaya conceived and designed the experiments, authored or reviewed drafts of the paper, and approved the final draft.

Zeti-Azura Mohamed-Hussein conceived and designed the experiments, authored or reviewed drafts of the paper, and approved the final draft.

The following information was supplied regarding data availability:

The 127 potential GSL genes and the 148 significant clusters are available in the Supplemental Files.

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
