# Peer review of "Potential Arabidopsis thaliana glucosinolate genes identified from the co-expression modules using graph clustering approach"

_PeerJ, doi:10.7717/peerj.11876_

## Round 0.1 · original submission · Minor Revisions

The two external reviewers agree on the merits of the manuscript, but also highlight a set of minor revisions that will improve the overall quality. I hope you will be able to address these issues adequately and provide me with an updated version and an overview of the changes you made in response to the reviewers. Looking forward to your new version.

Reviewer 1 ·

Basic reporting

Overall, the reporting is clear. However, there are some sentences that require rephrasing. Here are some specific examples:
-Line 65: “Glucosinolates (GSLs) are nitrogen-containing compounds” the “nitrogen containing” is redundant, please delete.
-Lines 251-252: “We hypothesize that co-exist genes with known GSL genes in the same statistically significant clusters can be used to predict potential GSL genes” – this sentence is not clear. Please rephrase/ explain.
-Lines 328-330: “Sulfur content in GSLs indicates that sulfur in plant is critical in GSL biosynthesis. Several metabolomic and transcriptomic studies reported a significant reduction in GSL accumulation under sulfur deficiency environment, suggesting the role of sulfur in GSL biosynthesis”- redundant, please rephrase.
-Lines 357-358: “In GSL biosynthesis, several groups of GSLs differ from the side chain derived from the amino acids that serve as the precursor” – rephrase.

Experimental design

The experimental design makes sense and is summarized very nicely in Figure 1. However, the reader will benefit if the authors try to engage Figure 1 to the results/methods sections by referring the reader to this figure throughout the manuscript, and by adding the relevant figure references to the steps describes in Figure 1.

Validity of the findings

The findings are well validated. No other comments.

Additional comments

Line 253-254: “Five sets of clusters were generated using density values of 0.5, 0.6, 0.7, 0.8, and 0.9 with cp value of 0.5”- Please elaborate, mainly on the density values and cp value, as part of the results, not only in the methods section, to give your results more context.
-Line 269: “Based on Table 3, the top six highly significant clusters … were chosen” – based on table S2, right? Table 3 is the six clusters.
-Figure 1 is very helpful in facilitating your workflow. Try to engage it to the results/methods sections by referring the reader to this figure throughout the manuscript, and by adding the relevant figure references to the steps describes in Figure 1.

Reviewer 2 ·

Basic reporting

The authors’ provide sufficient field background/context to understand the topic and purpose of the research. Although most of the relevant references are included, the authors’ overlooked an important publication, Wisecaver et al., (2017) The Plant Cell. Wisecaver et al. used an alternative method of gene co-expression analysis to identify specialized metabolite genes, in particular glucosinolate genes, in Arabidopsis thaliana and Brassica rapa. Some potential glucosinolate gene candidates recovered in this manuscript were identified by Wisecaver et al. previously. MRSA2 (AT5G07460) and AT1G21440 were recovered as co-expressed with glucosinolate genes in Wisecaver et al., supporting these genes may be involved in glucosinolate biosynthesis but not as novel candidates. This manuscript should consider referencing Wisecaver et al. and incorporate the findings from that study into the results, discussion, and conclusions of this manuscript.

Although the underlying data used in the study is already publicly accessible through well curated databases, the addition of the original gene co-expression matrix that was generated from these data sources, used in analyses, and referenced on lines 148-150 would further strengthen the quality and robustness of this manuscript.

Experimental design

No comment

Validity of the findings

No comment

Additional comments

The authors' have prepared a well-written manuscript reporting the results of their method of identifying glucosinolate biosynthesis genes and candidates. Identification of specialized metabolite genes through sequence similarity is problematic, and gene co-expression analysis have been shown to successfully recover clusters of genes involved in specialized metabolic pathways. There are many ways of performing gene co-expression analysis and results being dependent on the input gene expression data. This study provides an additional, alternative method of performing gene co-expression analysis to recover glucosinolate genes from the dataset, as well as identify additional glucosinolate gene candidates.

A small factual error stated should be corrected. Line 68 states “GSLs are exclusively found in the Brassicaceae family…”, which isn’t quite correct. Glucosinolates are predominantly produced by Brassicaceae species, but have been identified in other plants outside the Brassicaceae family.

---

## Round 0.2 · accepted · Accept

I have reviewed your responses to the critical, but minor, comments from the reviewers. To my mind, you have adequately addressed the issues raised, which makes the manuscript suitable for publication.